# Synthesis of Turbostratic Graphene with Micron-Sized Domains from Activated Charcoal by Fast Joule Heating

**DOI:** 10.3390/nano15241885

**Published:** 2025-12-15

**Authors:** Aisen Ruslanovich Prokopiev, Nikolay Nikolaevich Loskin, Pavel Vasilievich Vinokurov

**Affiliations:** Laboratory “Design-Center of Electronics «Sever»”, North-Eastern Federal University, 677000 Yakutsk, Russia; loskinnn@s-vfu.ru (N.N.L.);

**Keywords:** turbostratic graphene, fast Joule heating, activated charcoal, XRD, Raman spectroscopy

## Abstract

The development of economical and scalable methods for synthesizing high-quality graphene remains a pivotal challenge in materials science. This study presents an efficient approach for synthesizing turbostratic graphene with micron-sized domains from an accessible bioprecursor-activated charcoal—using fast Joule heating. We demonstrate that ultra-rapid thermal annealing (~16.2 kJ/g, up to 3000 K) triggers a phase transition from amorphous carbon to a highly graphitized structure. Comprehensive characterization via SEM, AFM, Raman spectroscopy, and XRD revealed the formation of large flakes with lateral dimensions up to 1.5 µm and thicknesses ranging from 4 to 200 nm. Raman mapping further uncovered a heterogeneous structure with alternating regions exhibiting different degrees of interlayer coupling, characteristic of turbostratic stacking. The key feature of the material is its turbostratic layer stacking, confirmed by the combination of XRD data showing an interlayer distance of 3.436 Å and Raman spectra characteristic of decoupled graphene layers. The synthesized material exhibits excellent electrical transport properties, with a bulk resistivity of 0.51 Ω·cm—an order of magnitude lower than that of the initial charcoal. These findings highlight the potential of the developed method for producing electrode materials for energy storage devices and conductive composites.

## 1. Introduction

Graphene, a 2D modification of the allotrope of carbon-graphite, is of considerable interest to the scientific community and manufacturers [1,2,3]. Its physicochemical properties have a number of advantages, such as zero band gap, mobility at the level of metallic bonds, and the possibility of modifying the structure without significantly disturbing the crystal lattice [4,5]. Over the past decade, ongoing research has led to the conclusion that the material and its derivatives (in particular, porous carbon materials) are suitable for the creation and development of more efficient batteries and supercapacitors based on it, as well as transparent conductive films [6,7,8]. Despite the above advantages, due to the complexity of obtaining graphene with monolayer properties (mechanical exfoliation of graphite), there is still no method for synthesizing graphene on an industrial scale, which, in general, hinders the introduction of this material into industrial production. Of course, there are various physical and chemical methods for obtaining graphene, in particular, chemical/plasma chemical vapor deposition (CVD/PECVD) and laser-irradiation method which can be used to obtain graphene on an industrial scale [7,9,10]. However, these synthesis processes are characterized by significant energy and economic costs, the complexity of controlling the growth of graphene flakes, and the fact that the grown domains have low lateral dimensions [11].

The solution to this problem is turbostratic graphene—a multilayer structure in which adjacent graphene layers are rotated relative to each other by a random angle [12]. This configuration weakens the interlayer interaction, allowing the material to retain many of the desirable electronic properties of a monolayer, while making it significantly easier and cheaper to produce in large volumes [13,14].

The relevance of turbostratic graphene for industry is confirmed by a large number of recent studies. The authors of [13] emphasize that this material is ideal for bulk synthesis and use in energy storage devices. Studies also demonstrate the successful use of cheap and renewable carbon sources as raw materials, including biomass (sugar cane stalks, rice husks, leaves), plastic waste, and even food waste [14,15,16]. This not only reduces costs but also solves environmental problems. The study in [17] demonstrates that the introduction of small amounts of turbostratic graphene (0.001–10 wt.%) into materials such as concrete, plastics, and asphalt significantly improves their mechanical and conductive properties [17].

In recent years, significant interest has been drawn to rapid or pulsed heating methods that enable the synthesis of turbostratic graphene from solid-phase carbon precursors under scalable conditions. Among the industrially promising approaches, the following can be distinguished: (1) flash Joule heating (FJH) [15,16,18], (2) long-pulse Joule heating (LPJH) [14], (3) high-temperature graphitization of carbon materials by plasma spraying at ~3500 K [13], (4) CVD variants leading to the formation of multilayer turbostratic structures [18,19], as well as (5) laser-assisted methods [20,21].

Over the past two years, a number of studies have been published demonstrating the extended capabilities of FJH and related methods for producing highly misoriented carbon structures. In particular, the work of Luong et al. [15,16,18,22] and research from 2023–2025 on FJH, LPJH, and carbothermal shock indicate that ultrafast heating (>2000–3000 K) followed by rapid quenching promotes the formation of misoriented layers, reduced interlayer coherence, and expanded interlayer spacing (d_002_ > 0.34 nm). Nevertheless, most previously described approaches have yielded relatively small lateral domain sizes (typically 50–300 nm), leaving open the question of the feasibility of obtaining larger, micron-sized turbostratic domains.

Based on the literature review, it can be concluded that the most advanced and economically viable strategy for obtaining turbostratic graphene is a combination of two elements: the use of cheap and readily available carbon raw materials and the application of the Joule heating method for its conversion into turbostratic graphene. Activated charcoal, a renewable resource of biomass origin, was selected as a promising precursor. The use of such carbon materials not only reduces the cost of the final product, but also corresponds to current trends in the utilization of carbon-containing waste [23]. Thus, the choice in this work of activated charcoal as a precursor is due to its widespread availability and high specific surface area (from 800 m^2^/g), which creates optimal conditions for the formation of large graphene domains during fast Joule heating.

Herein, we present a method for obtaining and studying high-quality turbostratic graphene by exposing activated charcoal to fast Joule heating (fJH). Previously, graphene-containing powders were obtained from plastic waste using the author’s method of fJH, in particular, turbostratic graphene [24], which has physicochemical properties similar to those of graphene, as well as rapid and environmentally friendly synthesis. The results obtained open up prospects for the creation of cost-effective and scalable technologies for processing biomass waste into high-quality carbon materials with a controlled structure.

## 2. Materials and Methods

Commercially produced crushed activated charcoal of the BAU-A grade (Perm Sorbent Plant UralkhimSorb LLC, Perm, Russia) with particle sizes ranging from 1.0 to 3.6 mm and a specific surface area of ~800 m^2^/g was used as raw material. The activated charcoal was exposed to fJH without pretreatment using a custom-designed setup. A detailed description of the design and operating principle of the setup is given in [24]. The unit includes 5 capacitor blocks with a total capacity of 180 mF, charged to a voltage of 300 V. Detailed information about the setup is provided in Appendix A. The FJH parameters (300 V, 16.2 kJ/g) were selected based on optimized conditions previously established for turbostratic graphene synthesis from carbonaceous precursors [15]. While the specific precursor differs, the energy input and pulse duration were maintained to ensure similar thermal conditions (~3000 K) necessary for the graphitization of amorphous carbon while preserving the turbostratic stacking through rapid quenching [15]. A sample weighing 0.5 g was loaded into a quartz tube with an inner diameter of 10 mm and pressed on both sides with graphite electrodes. The prepared tube was placed in a vacuum chamber at a pressure of 0.3 bar, which was provided by a Value VE115N vacuum pump (Value, Hangzhou, China). A single high-voltage discharge, with a pulse duration of ~720 ms to ~1.8 s depending on the resistivity at the ends of the electrodes, provided a specific energy of 16.2 kJ/g, calculated using the capacitor discharge energy Formula (1):(1)$$E=\frac{CU^{2}}{2m}$$
where m is the sample mass, C is the capacitor capacity, and U is the capacitor block voltage. After discharging the fJH, the product was cooled naturally for ~3–5 min to room temperature. Figure 1 shows a brief diagram of the processes.

All samples were synthesized at a constant voltage of 300 ± 0.5 V, regulated by a digital voltmeter connected to an Arduino controller. The peak current reached approximately 75 A per sample during the discharge. The synthesis procedure was reproduced at least 11 times, and all experiments were successful, producing materials with consistent structural, Raman, and electrical properties as presented in this study (see Appendix A).

After exposure to fJH, the activated charcoal, which was originally black in color (Figure 1), swelled slightly and turned predominantly gray (Figure 1). The synthesized materials were investigated using the following methods and equipment: scanning electron microscopy (SEM, JEOL 7800F, Tokyo, Japan) with an energy-dispersive X-ray spectra (EDS, Oxford Instruments, Abingdon, UK), X-ray diffraction powder diffractometer (XRD, ARL X’TRA Thermo Fisher Scientific, Waltham, MA, USA), atomic force microscopy (AFM, Ntegra Spectra, Zelenograd, Russia), and Raman spectroscopy (Ntegra Spectra, Zelenograd, Russia). The method parameters:SEM: images were acquired at an accelerating voltage of 3 kV.AFM: Surface morphology measurements were performed in semi-contact (tapping) mode using standard NSG10 cantilevers (TipsNano, Zelenograd, Russia) with a probe tip radius of 10 nm.Raman spectroscopy: Spectra were acquired using a green laser with a wavelength of 532 nm (2.33 eV). The diameter of the focused laser beam was 0.5 µm.XRD: Spectra were acquired using following parameters: Radiation—Cu Kα (λ = 1.5406 Å); Operating conditions: 40 kV, 40 mA; Scan range: 3–60° 2*θ*; Scan rate: 2°/min; Step size: 0.02°.Correlative AFM-Raman analysis: AFM imaging in tapping mode was used to locate individual flakes and measure their thickness. Raman spectra were then acquired from the center of the same flakes using a 0.5 µm laser spot.

The electrical properties were investigated in two stages. In situ measurements were performed directly in the reaction chamber between graphite electrodes using a CEM DT-9985 multimeter (CEM, Shenzhen, China) with a resistivity measurement error of ±0.3%. Ten independent synthesis processes showed a decrease in resistivity from 4–6 Ω of the initial charcoal to 1–1.5 Ω after fJH. To evaluate the electrical conductivity of the synthesized turbostratic graphene, pellets were fabricated by rapid pressing of 0.5 g of powder under ~10 MPa. The pellets, with a diameter of 18 mm and a width of 5 mm, were formed using 2–3 drops of ethanol as a temporary binder. For comparison, pellets were also made from the initial activated charcoal under the same conditions and formation parameters. All samples were pressed for 5 min. The bulk resistivity (ρ) of the compressed samples was measured using a four-probe method at room temperature and controlled humidity, and calculated using Formula (2) considering the geometric parameters of the pellet:(2)ρ=R×S/l
where ρ is the bulk resistivity of the material, Ω·cm, R is the measured resistivity, Ω, l is the length, cm, S is the cross-sectional area, cm^2^.

The measurements were performed on an AKTAKOM AMM-3046 component analyzer (AKTAKOM, Moscow, Russia) at a frequency of 50 Hz and with a basic measurement error of ±0.05%, performing 11 measurements for each pellet.

## 3. Results

### 3.1. Morphological and Structural Analysis

The morphology of the initial and modified materials was studied using scanning electron microscopy. The initial activated charcoal exhibits a heterogeneous loose structure with small aggregated particles of irregular shape (Figure 2a). Bright spots of charge carrier accumulation and charging are observed on the surface, which are characteristic of dielectric materials.

After fast Joule heating, a radical transformation of the morphology of activated charcoal occurs (Figure 2b). The formation of flat lamellar particles (flakes) with clearly defined boundaries is observed. The image shows both individual flakes and their large agglomerates. The lateral dimensions of isolated flakes, estimated from SEM micrographs, range from 0.5 μm. The inset to Figure 2b presents an enlarged view of an individual flakes, where a distinct plate-like morphology with well-defined, angular boundaries can be observed.

### 3.2. Elemental Composition

The elemental composition of materials in the general area of the SEM images and point analyses in the formed domains was investigated using EDS. The obtained values were averaged using the arithmetic mean method. The results of the quantitative analysis are presented in Table 1.

After fast Joule heating, a statistically significant change in the elemental composition is observed. The carbon content increases from 95.0 at. % to 98.5 at. %, while the oxygen content decreases from 5.0 at. % to 1.5 at. %. A key indicator is a sharp decrease in the O/C ratio from 0.052 to 0.015 (2.5 times). This indicates that during the Joule heating process, significant graphitization and removal of oxygen-containing functional groups occurred. It should be noted that oxygen was completely absent in some areas. Parallel to the purification of the material from oxygen-containing groups, there is a decrease in electrical resistivity from 4–6 Ω to 1–1.5 Ω.

### 3.3. X-Ray Diffraction Analysis

XRD (Figure 3) confirms a fundamental change in the structure of activated charcoal as a result of fJH. The source material exhibits a diffraction profile typical of amorphous carbon with wide reflections in the range of ~22° and ~42°, which indicates a high degree of structural disorder [25]. Raw-data XRD-spectra are included in Appendix A.

After fJH, a radical structural restructuring is observed: the amorphous halo disappears, and clear, narrow peaks corresponding to the crystalline graphite-like phase appear. Peaks of (002) at 25.96°, (100) at 43.5° and (004) at 54° were recorded. The presence of reflex (004) is important evidence of the formation of a multilayered ordered structure along the “c” axis. A comparison with literature values is presented in Table 2.

The interlayer distance d_002_, calculated using the Bragg’s law formula for peak (002), is 3.436 Å. This value significantly exceeds that for ideal crystalline graphite (~3.35 Å) and is typical for turbostratic graphene, in which neighboring layers are randomly rotated relative to each other, which weakens interlayer interactions and increases the distance [28].

The crystallite size along the “*c*” axis ($L_{c}$), estimated from the peak broadening (002) using the Scherrer equation:(3)$$L_{c}=(K\times \lambda )/(\beta \times cos\theta )$$
where: K = 0.9 (Scherrer constant for graphene), λ = 1.5406 Å (wavelength of Cu Kα radiation), β = FWHM in radians (width at half height), θ = peak position in radians (*θ* = 2*θ*/2).

It was approximately 6.2 nm. Considering the interlayer distance, this corresponds to ~18 graphene layers in the most ordered domains. It is important to note that the $L_{c}$ parameter, determined by the broadening of the reflex (002) using the Scherrer equation, characterizes not the total thickness of the particles, but the average size of the coherent scattering domains along the c axis, that is, the average height of the layer packages with an ideal stacking periodicity [29]. In turbostratic carbon materials, where neighboring graphene layers are randomly rotated relative to each other, it is these rotational stacking defects (misorientation) that limit the region of coherent X-ray scattering and, accordingly, the *L_c_* value, which can be significantly less than the total morphological thickness of the particle.

The totality of the X-ray diffraction data—namely, the increased interlayer distance, the presence of reflexes (004) and the final size of the crystallites—clearly indicates the formation of a turbostratic rather than Bernal (AB-stacking) graphite structure. The obtained parameters (d_002_, $L_{c}$) are in good agreement with the literature data for turbostratic graphene synthesized by flash Joule heating methods [13,14,15].

### 3.4. Surface Topography and Flake Thickness

AFM images show the presence of distinct lamellar particles with hexagonal morphology (Figure 4a). Statistical analysis of particle sizes based on AFM micrographs shows that the lateral dimensions of the flakes vary in the range from >0.1 to ≥1 μm. The observed particle shape indicates the crystalline nature of the material and corresponds to the thermodynamically stable configuration of graphene structures [30].

Particle thickness measurements were performed by analyzing the height profile at the flake-substrate interface. The results show that the thickness of the synthesized flakes ranges from a few nanometers to 200 nm (Figure 4b–d). It is important to note that this morphological thickness measured by AFM represents the total physical dimension of the particles, whereas the crystallite size along the c-axis ($L_{c}$ ≈ 6.2 nm) obtained from XRD analysis (Section 3.3) characterizes the average height of coherently scattering domains with regular interlayer spacing. In turbostratic carbon materials, the random rotational misorientation between adjacent graphene layers limits the coherence length of X-ray scattering, resulting in $L_{c}$ values that are typically smaller than the total particle thickness observed by AFM [29].

A combined analysis of lateral dimensions and thickness revealed a clear trend: the largest domains (with lateral dimensions of 0.8 μm) also have the greatest thickness (up to 200 nm), while small particles (~100–200 nm) are significantly thinner (4–10 nm). The formation of such elongated and thick crystallites is unusual for many methods of graphene material synthesis and indicates a specific growth mechanism under conditions of fast Joule heating. In addition, the presence of flat terraces with stepped boundaries (Figure 4a) is direct evidence of the layered crystalline structure of the material and visually confirms its graphene nature.

Despite the significant difference in particle thickness, phase contrast analysis did not reveal any pronounced structural defects within individual flakes. The uniform distribution of the phase signal across the particle surface indicates a high degree of crystallinity of the material and the absence of significant disorder in the packing of carbon layers.

### 3.5. Raman Spectroscopy

The results of Raman spectroscopy studies revealed a fundamental structural transformation of amorphous activated charcoal into highly ordered graphene nanostructures under the influence of fJH. As shown in Figure 5a, the initial material exhibited typical Raman spectra characteristic of amorphous carbon: a broad and intense D-band (~1350 cm^−1^), indicating a highly disordered structure and the presence of defects, and a broad G-band (~1580 cm^−1^), reflecting the vibrations of sp^2^-hybridized carbon atoms. The intensity ratio *I_D_*/*I_G_* significantly exceeded unity, which is typical for highly defective carbon materials. A broad D + G band, characteristic of amorphous carbon materials containing small nanographite flakes, was also observed [31,32].

Figure 5b–d presents typical Raman spectra of the synthesized material, taken at different points on the sample. All spectra show a characteristic triplet structure consisting of the D-band (~1350 cm^−1^), G-band (~1580 cm^−1^), and 2D-band (~2700 cm^−1^). It is also important to note that these peaks were fitted with a single Lorentzian function, which is characteristic of the highest quality graphene structures. The lateral domain sizes ($L_{a}$) were calculated according to the quantitative model proposed by Cançado et al. [33]. The presence of large domains correlates with SEM and AFM data. The highest recorded *I_D_*/*I_G_* ratio of ≥0.15, registered in the area in Figure 5d, indicates the formation of extended graphene domains with sizes exceeding 100 nm.

The analysis of the 2D-band is most informative for determining the number of layers and stacking type in graphene structures. In areas with the most ordered structure (Figure 5b), a narrow symmetric 2D-band (~2692 cm^−1^, FWHM = 20 cm^−1^) with an intensity ratio *I*_2*D*_/*I_G_* ≥ 2.5 is recorded, which in the literature is usually associated with monolayer graphene [34]. The G-band parameters, as well as the D-band at the detection limit, also indicate high crystallinity of the material. The FWHM of the G-band is 16 cm^−1^, which is characteristic of monolayer graphene. The position of the G-band at ~1581 cm^−1^ indicates the absence of significant mechanical stress in the carbon plane. However, as will be shown later, despite the apparent formation of monolayer graphene on the surface of the synthesized material, the contradiction with AFM results (tens and hundreds of graphene layers) is resolved by the formation of a different graphene structure after fJH.

In the Raman spectra shown in Figure 5c, a weak D-band (1338 cm^−1^ FWHM = 26 cm^−1^) with a ratio *I_D_*/*I_G_* ≈ 0.04 is registered. This minimal value, at the detection limit, also indicates a low defect density, comparable to graphene obtained by mechanical exfoliation. The G-band is also sharp (~1576 cm^−1^, FWHM = 22 cm^−1^) and indicates high crystallinity. The 2D-band (~2691 cm^−1^, FWHM = 31 cm^−1^) retains a symmetric shape, but its intensity becomes approximately equal to that of the G-band (*I*_2*D*_/*I_G_* ≈ 1.54), which is a characteristic feature of bilayer graphene in the literature [34]. The $L_{a}$ values estimated by the empirical formula proposed by Cançado et al. [33] reached up to 457 nm:(4)$$L_{a}=(2.4\;\times \;10^{-10})\lambda^{4}{\left(\frac{I_{D}}{I_{G}}\right)}^{-1}$$
where $L_{a}$—lateral size sp^2^-crystallites, nm; λ—laser beam length, nm; $\frac{I_{D}}{I_{G}}$—*I_D_*/*I_G_* ratio.

In the Raman spectra shown in Figure 5d, a weak D-band (~1346 cm^−1^, FWHM = 56 cm^−1^) is detected. A narrow and highly intense G-band (29 cm^−1^ FWHM) is also observed at ~1572 cm^−1^. The integrated intensity ratio of the D and G bands is ~0.15, which also indicates that the formed structure possesses a high degree of structural perfection. A 2D-band (~2696 cm^−1^, FWHM = 45 cm^−1^) with an intensity equal to about half of the G-band is also detected. The ratio was *I*_2*D*_/*I_G_* ≈ 0.76, which is a sign of few-layer graphene [35]. The $L_{a}$ values calculated from the formula were ~128 nm, which is consistent with the laser beam diameter. Parameters of turbostratic graphene obtained by AFM, SEM, and Raman spectroscopy are presented in Table 3.

However, as mentioned above, the domain thicknesses range from a few to 200 nm and above, corresponding to ~12 to nearly 600 graphene layers. This contradiction is resolved within the model of turbostratic graphene [27,36]. Turbostratic stacking, characterized by a random rotation angle between adjacent graphene layers, leads to a weakening of the interlayer interaction. As a result, the Raman spectra of such multilayer structures exhibit properties inherent to monolayer/bilayer graphene—a narrow symmetric 2D-band with a high *I*_2*D*_/*I_G_* ratio [15,17].

Thus, the observed combination of particle thicknesses (predominantly from ~12 layers up to ~600) with Raman spectral characteristics typical of monolayer and few-layer graphene (*I*_2*D*_/*I_G_* ≥ 2.5, symmetric, see Appendix A) is explained by the weakened interlayer interaction inherent to turbostratic stacking [15,36]. The variation in *I*_2*D*_/*I_G_* ratios across the sample surface reflects heterogeneity in the angular orientation of adjacent layers, rather than a difference in their number.

#### Spatial Mapping of Structural Properties

A comprehensive analysis of the spatial distribution of Raman parameters (Figure 6) reveals a consistent domain structure of the synthesized turbostratic graphene. A clear correlation is observed between three key parameters: areas with high G-band intensity (ordered sp^2^-domains) correspond to zones with a high *I_D_*/*I_G_* ratio and minimal D-band intensity (low defect density) [34,36,37]. The sizes of the identified domains reach >1 µm, confirming the formation of a macrocrystalline structure and exceeding most previously reported values for turbostratic graphene [14,15]. The mosaic distribution of parameters is characteristic of turbostratic graphene, where the random angular orientation of neighboring layers leads to spatial heterogeneity of electronic properties while maintaining high crystallinity within individual domains [38].

The ratio of the absolute intensity values *I_D_*/*I_G_* was ~0.8. Also, the *I*_2*D*_/*I_G_* ratios showed values of ~1.1–1.9, which in combination confirms the formation of a large-domain turbostratic structure with minimal defect density. At the same time, as established in Section 3.5, the *I_D_*/*I_G_* ratios were up to 0.2. The contradiction in the data is reconciled by the fact that the Raman mapping method integrates the signal from the entire scanning area, including not only ideal domains. Point spectra reflect the structural perfection of individual domains, while mapping characterizes the overall structural heterogeneity of the material.

Figure 7 shows a plot of the relationship between the geometric parameters of the carbon domains and the degree of structural defects. Small domains (≤200 nm, thickness ≤ 50 nm) are characterized by a higher concentration of defects, reflected in increased *I_D_*/*I_G_* values. With an increase in lateral and vertical dimensions (≥600 nm, ≥60 nm, respectively), the defect density decreases. The largest domains (≥1 µm, ≥200 nm in thickness) correspond to a more ordered graphite-like structure, indicating a clear trend towards improved crystallinity with increasing domain sizes.

Thus, the analysis of Raman spectra in combination with SEM and AFM allows us to conclude the formation of a highly crystalline graphene material with turbostratic layer stacking after fast Joule heating. The observed spectral characteristics indicate a minimal defect concentration and the presence of regions with weakened interlayer interaction, which explains the combination of “monolayer-like” Raman spectral characteristics with the significant material thickness established by AFM. The *I_D_*/*I_G_* distribution results confirm the observed correlation between morphological features and the degree of disruption of the crystal lattice in graphene domains with thicknesses from 4 to 200 nm.

The presented Raman mapping results (Section Spatial Mapping of Structural Properties) provide a visualization of the heterogeneous structure of the obtained turbostratic graphene. The distribution of Raman parameters is consistent with the model of turbostratic material stacking, where the random angular orientation of neighboring layers leads to spatial variation in electronic properties. Large domains (>1 µm) with low D-peak intensity and high *I*_2*D*_/*I_G_* ratios correspond to regions with minimal defect density and weakened interlayer interaction. The observed material heterogeneity is associated with the preferential occurrence of fJH in the most densely packed sp^2^-crystallites of the activated wood charcoal and is explained by the non-equilibrium propagation of the temperature field and their crystallization rates.

### 3.6. Electrical Transport Properties

The synthesized turbostratic graphene exhibits remarkable electrical properties. Pressed pellets show bulk resistivity of 0.51 ± 0.02 Ω·cm, which is 5–10 times lower than pellets made from initial charcoal (2.52–5.13 ± 0.02 Ω·cm).

The sharp decrease in bulk resistivity by an order of magnitude after fast Joule heating and the possibility of achieving high conductivity clearly indicate a fundamental change in the conductivity mechanism. This confirms the formation of a highly conductive metastable phase of turbostratic graphene, whose flakes effectively form a conductive network due to low interparticle contact resistivity and high conductivity.

Analysis of the electrical properties reveals a sequential change in the conduction mechanism. The initial amorphous charcoal is known to be characterized by percolation conduction via a hopping mechanism between closely spaced sp^2^-nanocrystallites (~8.5 nm) [38,39,40]. The reduction in resistance from 4–6 Ω to 1–1.5 Ω after fJH, measured under identical conditions, indicates a fundamental change in the conduction mechanism—a transition from hopping conduction in an amorphous matrix to efficient charge transport within a system of graphene domains. Most significant for practical application is the material’s behavior upon pressing. The achievement of a consistently low bulk resistivity of 0.51 ± 0.02 Ω·cm at a pressure of 10 MPa is a key result. It demonstrates both the minimization of interparticle contact resistance and the high intrinsic conductivity of the flakes. This performance is superior to that of most pressed bulk carbon materials reported via similar or more elaborate routes [41].

## 4. Discussion

The comprehensive analysis conducted allows us to establish that fJH induces a fundamental structural rearrangement of the amorphous activated charcoal, leading to the formation of large-domain turbostratic graphene. The most significant result of this work is the discovery of a strong correlation between the lateral size and thickness of the turbostratic graphene domains (Section 3.5, Figure 7), which is not typical for most literature data on turbostratic graphene synthesis. In contrast to most literature reports, where turbostratic graphene domain sizes do not exceed hundreds of nanometers [14,15,42], in our case, the formation of substantially larger domains with a lateral size of >1 µm is observed. This order-of-magnitude difference may be attributed to the specific features of the initial precursor (activated charcoal) and the optimized parameters of fJH.

The physical model of the transition from an amorphous state to a metastable, heterogeneous graphene structure, proposed below, explains all observed phenomena, including apparent contradictions between data from different characterization techniques.

### 4.1. Model of Structural Transformation

The transformation process involves three stages: (1) pyrolytic decomposition of oxygen-containing groups and melting of the amorphous activated charcoal matrix; (2) recrystallization with the formation of extended hexagonal planes; and (3) formation of a turbostratic structure with random angular orientation of the layers due to kinetic constraints during cooling. The initial activated charcoal, exhibiting bright charging artifacts in SEM data (white dots and spots), characteristic of dielectric and disordered systems [43], transforms into large, plate-like particles with hexagonal morphology. EDS data confirm the purification of the material (decrease in oxygen content from ~5.0 to ~1.5–1.0 at. %), while SEM and AFM demonstrate the formation of isolated flakes with lateral sizes >1 µm and thicknesses ranging from a few to 200 nm.

Further evidence for this model is provided by XRD data. The initial charcoal exhibits broad reflections typical of an amorphous state. After fJH, a radical structural rearrangement is observed: sharp (002), (100), and (004) reflections appear. A key argument is the interlayer distance d_002_ = 3.436 Å, which significantly exceeds that of ideal graphite (~3.35 Å) and is unambiguously characteristic of turbostratic graphene, where rotational misorientation between adjacent layers weakens their interlayer interactions [15,28,36]. Under conditions of ultra-fast heating and subsequent rapid cooling, the formed layers do not have time to align into a perfect Bernal-stacked graphite, forming a turbostratic structure with large lateral domains.

The observed growth of large, thick domains can be explained by the microstructural heterogeneity of the precursor. During fJH, current concentrates in the most conductive percolating networks of dense sp^2^-clusters, causing localized Joule overheating. This non-equilibrium thermal spike promotes rapid lateral and vertical growth within these regions, while less ordered areas experience milder heating. The subsequent ultrafast quenching freezes this structurally inhomogeneous state, preserving the large turbostratic domains formed in the overheated zones and accounting for the correlation between domain size and thickness (Figure 7).

### 4.2. Resolution of Key Contradictions Within the Turbostratic Graphene Model

The observed combination of significant particle thicknesses, ranging from ~12 to nearly 600 graphene layers (as per AFM) with Raman spectra characteristic of decoupled few-layer graphene (*I*_2*D*_/*I_G_* ≥ 2.5) is explained by the weakened interlayer interaction inherent to turbostratic stacking [12,13,15,44]. The variation in *I*_2*D*_/*I_G_* ratios across the sample surface reflects the heterogeneity in the angular orientation of adjacent layers, rather than a difference in their number.

### 4.3. Spatial Structural Inhomogeneity by Raman Mapping

The presented Raman mapping results (Section Spatial Mapping of Structural Properties) provide a picture of the heterogeneous structure of the obtained turbostratic graphene. The distribution of Raman spectroscopy parameters agrees with the model of turbostratic material stacking, where the random angular orientation of neighboring layers leads to spatial variation in electronic properties. Large domains >1 µm with low D-peak intensity and high *I*_2*D*_/*I_G_* ratios correspond to regions with minimal defect density and weakened interlayer interaction. The observed material inhomogeneity is associated with the fact that the fJH process initiates within the most densely packed sp^2^-crystallites of the wood-derived activated carbon and is explained by the non-equilibrium propagation of the temperature field and their crystallization rates.

### 4.4. Evolution of the Electrical Conductivity Mechanism

Analysis of the electrical properties demonstrates a fundamental evolution in the conduction pathway resulting from fast Joule heating. The initial material exhibits hopping conductivity between nanoscale sp^2^-domains, typical of amorphous carbon. Following fJH, a sharp reduction in resistance—from 4–6 Ω to 1–1.5 Ω under identical measurement conditions—signals a transition to efficient, domain-mediated charge transport. The most practically significant evidence of this transformation is the low bulk resistivity of 0.51 ± 0.02 Ω·cm achieved in pressed pellets at only 10 MPa. This outcome indicates that:A highly conductive percolating network forms with minimal processing intervention;The turbostratic graphene flakes possess a strong inherent tendency for efficient self-assembly and packing;Substantial interparticle contact is established even at relatively low compaction pressure.

The consistent attainment of such low resistivity under mild pressing confirms that the flakes exhibit high intrinsic conductivity and an exceptional ability to form coherent, low-resistance conductive pathways.

### 4.5. Correlation of Morphological Parameters

The discovered correlation between the lateral sizes of the domains and their thickness (~100 nm/4 nm vs. >1 µm/200 nm) indicates competition between lateral and vertical growth under kinetically limited recrystallization conditions, which further confirms the proposed physical model. Thus, the totality of experimental data convincingly proves the synthesis of turbostratic graphene, whose electrophysical properties are determined by both the high crystallinity of individual domains and the morphology, opening prospects for the creation of electrode materials and conductive composites.

### 4.6. Comparison of Turbostratic Graphene Production Methods

Table 4 provides a comparative analysis with the literature data on the production of turbostratic graphene by Joule heating methods. For comparison with our work, we selected the original Joule heating method presented in [15] utilizing various carbon precursors, and the more modern long-pulse Joule heating method for producing turbostratic graphene from biomass [14].

Although all three methods successfully produce turbostratic graphene without using solvents, demonstrating their common advantage of environmental sustainability, key differences in the characteristics of the obtained materials highlight the efficacy of our fJH process using activated charcoal. An important parameter characterizing graphene properties is the *I*_2*D*_/*I_G_* peak intensity ratio in Raman spectra. The value of this ratio (≥2.5) observed in our case is indicative of weakened interlayer interaction, typical of turbostratic stacking [15], and exceeds the value achieved by the DC-LPJH method.

The most substantial difference lies in the domain size of the obtained materials. The FJH method presented in [15] yields turbostratic graphene with domain sizes up to ~100 nm. Our fJH variant, applied to activated charcoal, produces substantially larger domains with a lateral size exceeding 1 µm and a more ordered graphitic structure.

From an energy consumption perspective, our fJH process requires more energy (~16.2 kJ/g) compared to the cited methods. However, this higher energy input appears to be a key factor enabling the formation of large domains through more extensive structural rearrangement of the material.

## 5. Conclusions

In this work, an efficient and environmentally friendly method for synthesizing high-quality turbostratic graphene from an accessible bioprecursor-activated charcoal—using high-temperature fast Joule heating (>3000 K) has been developed. A comprehensive investigation of the synthesized material by SEM, XRD, EDS, AFM, Raman spectroscopy, and electrical conductivity measurements established the following key results:This work demonstrates that fast Joule heating induces a fundamental structural rearrangement of amorphous carbon, leading to the formation of large hexagonal domains with sizes > 1 µm with thicknesses ranging from a few nanometers to 200 nm, while XRD revealed coherent domains of ~18 layers, confirming the turbostratic nature.The combined analysis of Raman spectroscopy and AFM data identified the material as turbostratic graphene. This is indicated by the combination of significant particle thickness and Raman spectral characteristics typical of few-layer graphene (minimal D-band intensity, narrow symmetric 2D-peak with *I_2D_*/*I_G_* > 2.5), which is explained by the weakened interlayer interaction due to chaotic angular orientation of the layers.The formation of a mosaic heterostructure with domain sizes > 1 µm was clearly demonstrated using Raman mapping.The resistivity of pressed pellets of the synthesized material was found to be as low as 0.51 Ω·cm. This indicates the formation of a dense conductive network with minimal contact resistance between individual graphene domains during pellet formation.

The obtained results demonstrate the promise of the developed method for creating electrode materials for supercapacitors and batteries, conductive fillers for composites, and other applications requiring high electrophysical characteristics combined with economic and environmental efficiency of production. The demonstrated fast Joule heating approach is inherently scalable, since both the voltage regulation system and the carbon precursor preparation can be easily adapted for multi-gram synthesis.

These results highlight the potential of this method for low-cost and sustainable production of turbostratic graphene from renewable carbon sources.

### Outlook for Future Work

The results obtained in this study form a foundation for a systematic research pathway aimed at advancing turbostratic graphene from fundamental understanding to practical implementation in energy storage devices. Future work will evolve along two complementary directions.

Fundamental Mechanisms, Parameter Mapping, and Structural Optimization.

Building upon the synthesis protocol established here, we will carry out a systematic investigation of the fJH parameter space—including energy input, pulse duration, and precursor density—to achieve controlled tuning of lateral domain size, layer number, and porosity (BET). Special emphasis will be placed on correlating these macroscopic parameters with the atomic-scale stacking order through high-resolution transmission electron microscopy (HR-TEM) and selected-area electron diffraction (SAED). These studies will complete the multi-scale structural picture of the material and clarify the conditions governing the formation of thick turbostratic domains.

2.Transition from Material to Device: Electrodes and Performance Evaluation.

The synthesized material will be further engineered into porous electrode architectures suitable for electrochemical energy storage. Detailed testing—including cyclic voltammetry, galvanostatic charge/discharge, and impedance spectroscopy—will be conducted to establish quantitative structure–property relationships and identify performance-limiting factors. Ultimately, this pathway aims at assembling and characterizing prototype supercapacitor devices based on this sustainable, large-domain carbon material.

Together, these research directions will provide a comprehensive understanding of both the formation mechanisms and the application potential of large-domain turbostratic graphene produced via fast Joule heating.

## 6. Patents

Prokopiev, A.R.; Matveev, V.I.; Loskin, N.N.; Popov, D.N. Method for Synthesis of Graphene-Containing Products from Polymer Materials. Russian Patent RU 2835422 C1, 25 February 2025.Prokopiev, A.R.; Matveev, V.I.; Loskin, N.N.; Popov, D.N. Device for Fast Joule Heating for Synthesis of Graphene-Containing Products from Polymer Materials. Russian Utility Model RU 229234 U1, 27 September 2024.

## Figures and Tables

**Figure 1 nanomaterials-15-01885-f001:**
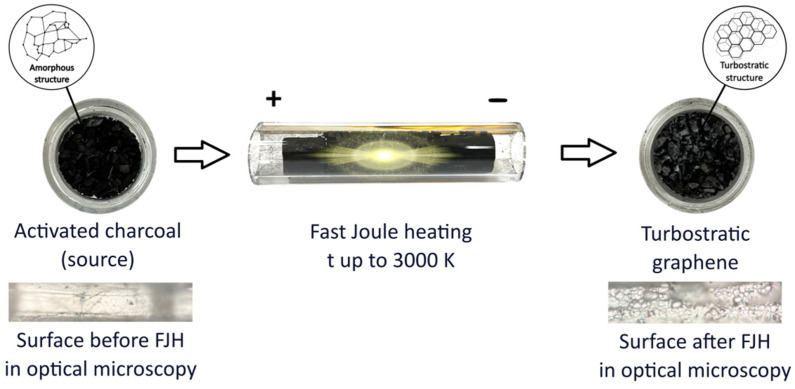
Schematic of the fast Joule heating process for the synthesis of turbostratic graphene from activated charcoal. Corresponding optical images of the sample before (black powder) and after (gray, expanded material) the discharge are shown below the schematic.

**Figure 2 nanomaterials-15-01885-f002:**
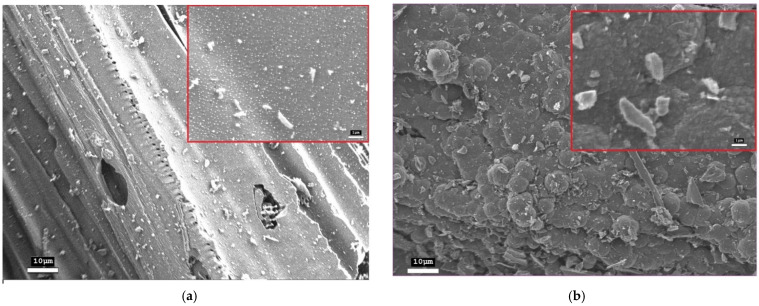
SEM images: (**a**) initial activated charcoal; (**b**) after fJH. The insets correspond to the SEM images in the center at a magnification of ×10,000 (ruler—1 μm).

**Figure 3 nanomaterials-15-01885-f003:**
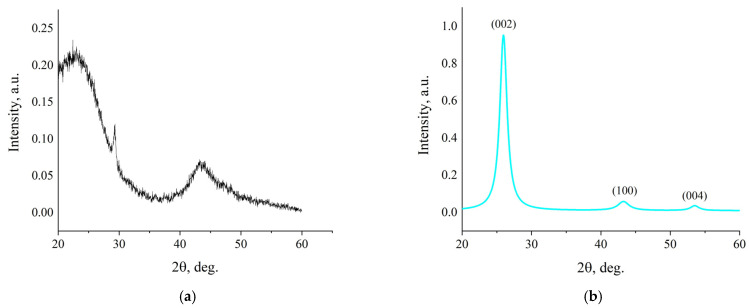
XRD spectra of: (**a**) initial activated charcoal; (**b**) activated charcoal after fJH.

**Figure 4 nanomaterials-15-01885-f004:**
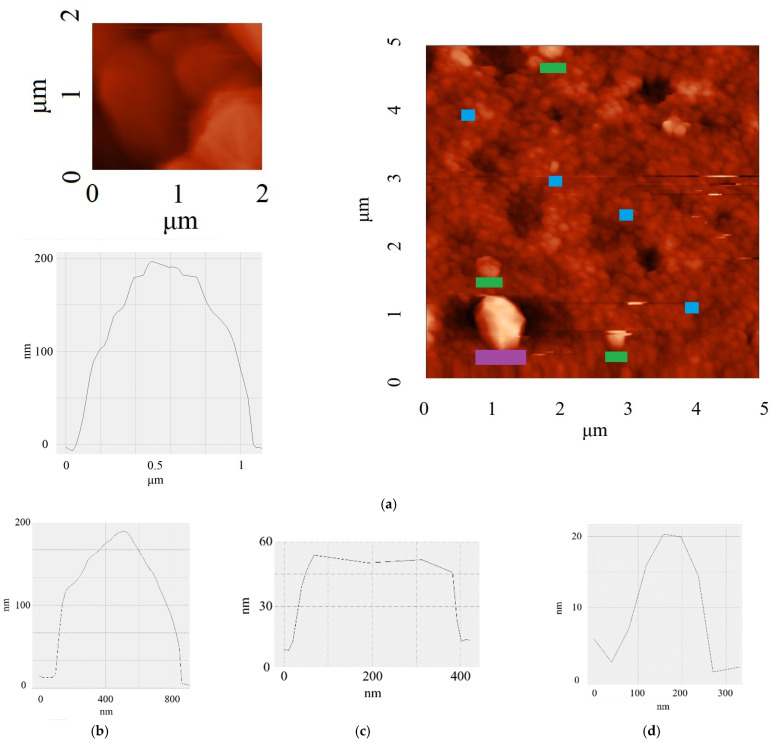
(**a**) AFM image 2 × 2 μm with a profile > 1 μm and 5 × 5 μm; profiles of 5 × 5 μm: (**b**) domain length 800 nm (purple); (**c**) domains up to 400 nm in length (green); (**d**) domains up to 200 nm long (blue).

**Figure 5 nanomaterials-15-01885-f005:**
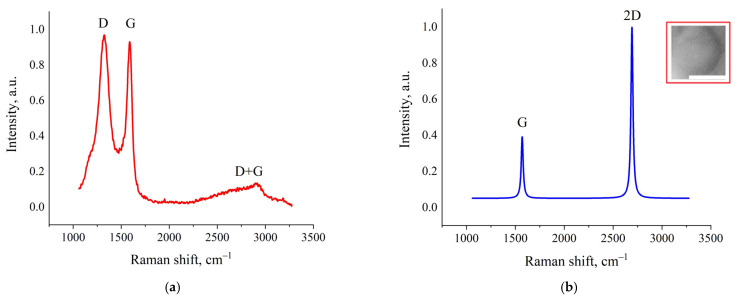
Raman spectra: (**a**) initial sample; Lorentzians fits: (**b**) post-fJH micron-sized regions; (**c**) post-fJH regions up to 800 nm; (**d**) post-fJH graphene sheets up to 400 nm. The insets show SEM images (presented in negative contrast for better visualization) containing graphene domains. The scale bars (white rectangles) correspond to 1 µm. Raw data of (**b**–**d**) are provided in the Appendix A.

**Figure 6 nanomaterials-15-01885-f006:**
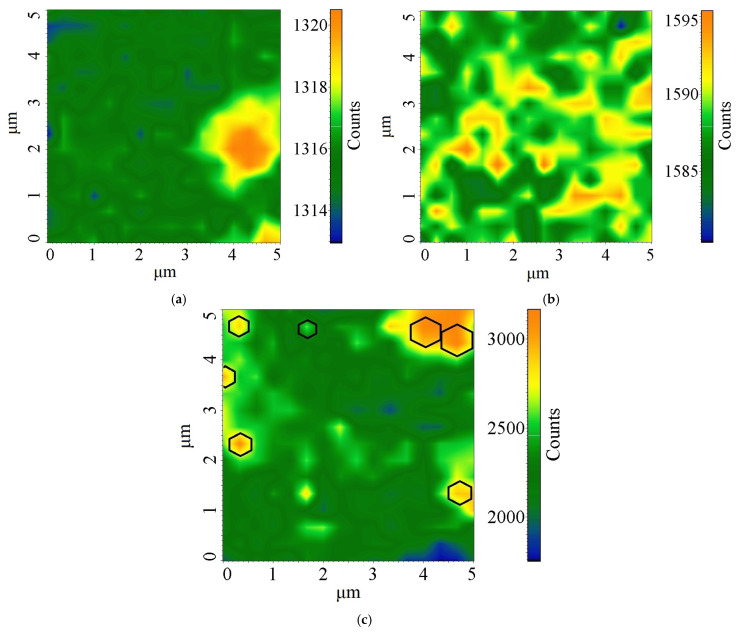
Correlation analysis of the structural parameters of turbostratic graphene using Raman mapping: (**a**) D-band intensity, (**b**) G-band intensity, (**c**) 2D-band intensity. The hexagons presented in figure for better visualization.

**Figure 7 nanomaterials-15-01885-f007:**
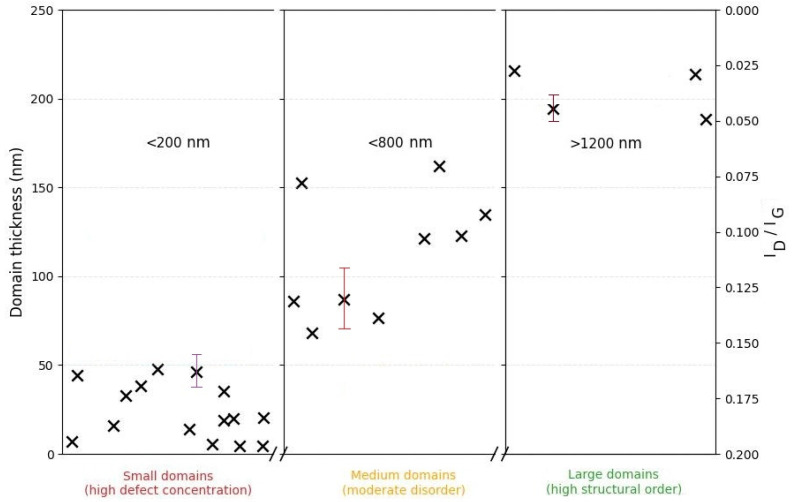
Correlation between the structural perfection (*I_D_*/*I_G_* ratio) and geometric parameters (lateral size and thickness) of the synthesized turbostratic graphene domains. The error bars (standard deviation) for domain thickness are: ±15 nm for domains ≤200 nm, ±40 nm for domains ≤800 nm, and ±8 nm for domains ≥1 µm.

**Table 1 nanomaterials-15-01885-t001:** Elemental composition of materials before and after fJH and in situ resistivity at the ends of electrodes before and after the reaction.

Samples	C, at. %	O, at. %	O/C	R (Ends of the Electrodes), Ω
Initial activated charcoal	~95.0 ± 0.5	~5.0 ± 0.5	0.052	~4–6 ± 0.2
Activated charcoal after fJH	~98.5 ± 0.5	~1.5 ± 0.5	0.015	~1–1.5 ± 0.2

**Table 2 nanomaterials-15-01885-t002:** Obtained data and comparison with literature values.

Parameter	Pristine Graphite (HOPG) [26]	Turbostratic Graphene [15,27]	This Work
(002) peak	~26.5° d = ~0.335 nm	Shifted downwards: 25.5–26.2° d = 0.340–0.349 nm	~25.95° Wide, intense
(100)/(101) peak	~42.3°	~42–43°	~42.72° (small, wide)
(004) peak	~54.5°	~53–54°	~53.96° (small)

**Table 3 nanomaterials-15-01885-t003:** Parameters of turbostratic graphene obtained by AFM, SEM, and Raman spectroscopy.

*L_a_*, AFM (µm)	*L_a_*, SEM (µm)	FWHM (G), cm^−1^	FWHM (2D), cm^−1^	*I_D_*/*I_G_*	*I*_2*D*_/*I_G_*	*L_a_*, Raman (µm)
≥1	≥1.5	16	20	n/d	≥2.5	-
≤0.8	≤0.7	22	31	≥0.04	≤1.5	≥0.45
≤0.2	≤0.3	29	45	≤0.2	≤0.8	≤0.13

**Table 4 nanomaterials-15-01885-t004:** Comparative analysis of turbostratic graphene and graphene-based materials production methods.

Carbon Precursor	Method	Energy/Power or Temperature	Raman (*I*_2*D*_/*I_G_*)	Interlayer (d_002_)	Domain Size	Eco Impact	Applications	Ref.
Charcoal	fast Joule heating	~16 kJ/g	≥2.5	3.44 Å	≥1 µm	Very Low (Green)	Energy Storage	This work
Various carbon sources (coal, petroleum coke, biochar, plastics, etc.)	Flash Joule heating	7.2–18 kJ/g	≤17	3.45 Å	>100 nm	Very Low (Green)	Composite materials	[15]
Biomass	Direct Current Long Pulse Joule Heating (DC-LPJH)	10 kJ/g	≤1.2	-	-	Low (Sustainable)	Composite materials	[14]
Biomass	Laser (LIG)	5–15 J/cm^2^	~1.1–1.5	~3.40 Å	~20–50 nm	Low (Sustainable)	Supercapacitors	[20]
Plastic Waste	AC and DC flash	~23 kJ/g	Up to 6	~3.45 Å	~10–60 nm	Very Low (Waste use)	Cement/Composites	[22]
Methane	PECVD	100–1000 W	D-peak present	Vertical	0.5–5 µm	High (Energy)	Sensing/Emission	[45]
Ethanol vapor	Direct CVD	~1573 K	~0.67–1.13	-	~60–100 nm	High (Energy)	Conductive Films	[19]

## Data Availability

The original contributions presented in this study are included in the article/Appendix A. Further inquiries can be directed to the corresponding author.

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
