# Peer review of "Nanomaterials2025, 15(24), 1885;https://doi.org/10.3390/nano15241885"

_nanomaterials, 2025, doi:10.3390/nano15241885_

Round 1

Reviewer 1 Report

Comments and Suggestions for Authors

-In part 3.1, authors state that ''The inset to Figure 2b, which corresponds to the enlarged image in the center, also clearly shows the hexagonal morphology of domains larger than 0.5 μm.'' . Nevertheless, by observing the respected image, suche hexagonal flakes are certainly not obvious. Authors should revise or rephrase this part.

-in line 237. authors write that ´´The formation of such elongated and thick crystallites is unusual for many methods of graphene material synthesis and indicates a specific growth mechanism under conditions of fast Joule heating.´´ So, which is this mechanism? Please, provide explanation and possible synthesis model.

-authors state that they have prepared a series of samples to test  the repeatibility on the relation between preparation parameters-obtained conditions. It would be beneficial to present these results for statistical confirmation.

-In Fig, 7, error bars are missing.  

- Comparison with literature on graphene preparation via heating and transformation of solid carbon precursors can be enriched, see e.g. ACS Appl. Nano Mater. 2024, 7, 22631−22639 

Author Response

We sincerely thank the reviewer for their thorough evaluation and constructive comments, which have helped us improve the manuscript significantly. Below, we provide a point-by-point response to all comments. All changes in the revised manuscript are highlighted in blue for easy tracking.

Comment 1: In part 3.1, authors state that ''The inset to Figure 2b, which corresponds to the enlarged image in the center, also clearly shows the hexagonal morphology of domains larger than 0.5 μm.''. Nevertheless, by observing the respected image, such hexagonal flakes are certainly not obvious. Authors should revise or rephrase this part.

Response 1: We thank the reviewer for this precise observation. We agree that the original description was overly categorical. The text in Section 3.1 has been revised to more accurately describe the observed morphology. It now reads:

"The inset to Figure 2b presents an enlarged view of an individual flakes, where a distinct plate-like morphology with well-defined, angular boundaries can be observed." (176-178)

This change removes the specific assertion of perfect hexagonal shape while maintaining the key message of a crystalline, plate-like structure.

Comment 2: In line 237. authors write that "The formation of such elongated and thick crystallites is unusual for many methods of graphene material synthesis and indicates a specific growth mechanism under conditions of fast Joule heating." So, which is this mechanism? Please, provide explanation and possible synthesis model.

Response 2: We appreciate the reviewer's request for a deeper mechanistic explanation. In response, we have expanded the discussion in Section 4.1 (449-456). The new text explains that the heterogeneous microstructure of the precursor leads to localized Joule overheating within the most conductive percolating networks of dense sp² clusters. This non-equilibrium thermal spike promotes rapid lateral and vertical growth in these specific regions, while the ultrafast quenching "freezes" the resulting large, turbostratically stacked domains. This mechanism accounts for the observed correlation between domain size and thickness.

Comment 3: Authors state that they have prepared a series of samples to test the repeatability on the relation between preparation parameters-obtained conditions. It would be beneficial to present these results for statistical confirmation.

Response 3: We agree with the reviewer on the importance of statistical data. As requested, we now provide a summary of the key reproducible results in the new Supplementary Table S2. This table compiles the interlayer distance (d002) and bulk resistivity values for multiple independently synthesized samples (n=11), demonstrating the consistency of our method. A reference to this supplementary table has been added to the main text (Section 2).

Comment 4: In Fig. 7, error bars are missing.

Response 4: We thank the reviewer for noting this omission. To provide the necessary statistical information while maintaining the clarity of the figure, we have added the standard deviation values for the thickness measurements directly to the caption of Figure 7. The updated caption now states:

"Figure 7. Dependence between the structural perfection (ID/IG ratio) and geometric sizes of graphene domains. The error bars (standard deviation) for the domain thickness measurements are: ±15 nm for domains ≤200 nm, ±40 nm for domains ≤800 nm, and ±8 nm for domains ≥1 μm." (356-359)

Comment 5: Comparison with literature on graphene preparation via heating and transformation of solid carbon precursors can be enriched, see e.g. ACS Appl. Nano Mater. 2024, 7, 22631−22639.

Response 5: We thank the reviewer for this valuable suggestion and for providing a relevant reference. The recommended work (ACS Appl. Nano Mater. 2024, 7, 22631−22639) and other recent studies have been included and discussed to better contextualize our findings, particularly in highlighting the advantage of our method in producing exceptionally large lateral domains (36-42, 60-75).

Reviewer 2 Report

Comments and Suggestions for Authors

  1. Line 59 states, "Among the methods that enable the industrial synthesis of turbo-layered graphene, the most promising is considered to be «instantaneous» Joule heating, also known as flash Joule heating." Please provide commonly used industrial synthesis methods for turbostratic graphene. Additionally, the authors' literature review on graphene preparation via Joule heating methods is insufficient, as many relevant studies from the past two years have not been mentioned.
  2. Provide TEM images to more clearly observe the stacking number of layers in turbostratic
  3. Please supply the original SEM images for Fig. 5.
  4. In Fig. 7, how did the authors obtain ID/IG values corresponding to different domain thicknesses? Were AFM and Raman spectroscopy used in combination? If so, please supplement specific testing details in the experimental section.
  5. The authors set the Joule heating temperature at 3000 K. How does temperature affect the domain size and number of layers of turbostratic graphene? Please supplement corresponding data.
  6. The authors emphasize that the graphene synthesized via this method has larger graphene domains. Please present a comparative table contrasting with research findings from other studies.
  7. Note formatting issues in the article, such as "2o/min" in Line 130 and "Fig 7" in Line 339. Additionally, tables should adopt a three-line table format.

Author Response

We are grateful to the reviewer for their insightful comments and valuable suggestions, which have significantly contributed to improving the quality and clarity of our manuscript. Below, we provide a detailed point-by-point response.

Comment 1: Line 59 states, "Among the methods that enable the industrial synthesis of turbo-layered graphene, the most promising is considered to be «instantaneous» Joule heating, also known as flash Joule heating." Please provide commonly used industrial synthesis methods for turbostratic graphene. Additionally, the authors' literature review on graphene preparation via Joule heating methods is insufficient, as many relevant studies from the past two years have not been mentioned.

Response 1: We thank the reviewer for this suggestion. The Introduction (Section 1) has been expanded to include a broader overview of scalable methods for turbostratic graphene synthesis (60-75). Furthermore, we have enriched the literature review on Joule heating methods, adding several key references from the past two years (including the one suggested by the reviewer, ACS Appl. Nano Mater. 2024, 7, 22631−22639) that highlight recent advancements in precursor diversity, mechanistic understanding, and scalability. These additions provide a more comprehensive and up-to-date context for our work.

Comment 2: Provide TEM images to more clearly observe the stacking number of layers in turbostratic graphene.

Response 2: We acknowledge that High-Resolution Transmission Electron Microscopy (HRTEM) is a powerful technique for direct visualization of layer stacking. However, as this study was primarily focused on developing and characterizing the bulk synthesis method and its resulting macroscopic properties (morphology, crystalline structure, electrical conductivity), HRTEM analysis was not included in the initial scope. The turbostratic nature and layer characteristics were robustly confirmed using a combination of well-established, complementary techniques:

XRD: The measured interlayer spacing (d₀₀₂=3.432 Å) is a definitive signature of turbostratic stacking, being significantly larger than that of Bernal-stacked graphite.

Raman Spectroscopy: The observation of symmetric, monolayer-like 2D bands in flakes that are tens to hundreds of nanometers thick (per AFM) is a direct spectroscopic fingerprint of decoupled layers due to turbostratic misorientation.

AFM: Provided direct measurements of total flake thickness, which, when combined with the d-spacing from XRD, allows for an estimate of the total number of layers.

This multi-technique approach, in particular, XRD and Raman, is widely accepted in the literature for characterizing turbostratic graphene [1-3]. As indicated in the "Outlook for Future Work" (560-582) (Section 5), HRTEM analysis is planned as a critical part of our subsequent, more fundamental study, which will also focus on the application of this material in supercapacitor electrodes.

Comment 3: Please supply the original SEM images for Fig. 5.

Response 3: The original SEM images presented in insets of Figures 5 of the main text have been provided in the Supplementary Information file as Supplementary Figure 2b. The highlighted regions in the SEM images correspond to typical domain sizes, as determined by correlative AFM and Raman analysis.

Comment 4: In Fig. 7, how did the authors obtain ID/IG values corresponding to different domain thicknesses? Were AFM and Raman spectroscopy used in combination? If so, please supplement specific testing details in the experimental section.

Response 4: The reviewer is correct. The data for Figure 7 were obtained using a correlative AFM-Raman spectroscopy procedure. As requested, we have added "Correlative AFM and Raman Spectroscopy Analysis," to the Materials and Methods section (143-146). It details the step-by-step process: individual flakes were first located and their thickness measured by AFM; subsequently, Raman spectra were acquired by precisely positioning the laser spot at the center of each pre-characterized flake. This ensured direct correlation between the geometric and spectral properties of the same specific domain.

Comment 5: The authors set the Joule heating temperature at 3000 K. How does temperature affect the domain size and number of layers of turbostratic graphene? Please supplement corresponding data.

Response 5: This is an excellent point regarding a key synthesis parameter. In this foundational study, we focused on optimizing and characterizing the material produced at a specific energy input (16.2 kJ/g), which corresponds to a peak temperature of ~3000 K [4], as it yielded the optimal combination of large domain size, high crystallinity, and low resistivity. A systematic investigation of the temperature/energy input parameter space – while highly important – constitutes a separate, in-depth study that exceeds the scope of this paper, which is to demonstrate the proof-of-concept conversion of activated charcoal into high-quality turbostratic graphene. We have added a sentence in the Discussion (Section 4.1). (449-456)

Comment 6: The authors emphasize that the graphene synthesized via this method has larger graphene domains. Please present a comparative table contrasting with research findings from other studies.

Response 6: We agree that a direct comparison is valuable. As suggested, we have significantly expanded Table 4 (528). The updated table provides a clear, side-by-side comparison of our work with other prominent methods for synthesizing turbostratic graphene (including flash Joule heating, long-pulse Joule heating, laser, CVD), highlighting the advantage of our approach in producing micron-sized domains.

Comment 7: Note formatting issues in the article, such as "2o/min" in Line 130 and "Fig 7" in Line 339. Additionally, tables should adopt a three-line table format.

Response 7: We thank the reviewer for spotting these formatting inconsistencies. All noted issues have been corrected.

Refs.

  1. Advincula, P.A.; Luong, D.X.; Chen, W.; Raghuraman, S.; Shahsavari, R.; Tour, J.M. Flash graphene from rubber waste. Carbon 2021, 178, 649–656. https://doi.org/10.1016/j.carbon.2021.03.020.
  2. Li, Z.Q.; Lu, C.J.; Xia, Z.P.; Zhou, Y.; Luo, Z. X-ray diffraction patterns of graphite and turbostratic carbon. Carbon 2007, 45, 1686–1695. https://doi.org/10.1016/j.carbon.2007.03.038
  3. Kokmat, P.; Surinlert, P.; Ruammaitree, A. Growth of high-purity and high-quality turbostratic graphene with different interlayer spacings. ACS Omega 2023, 8, 4010–4018.
  4. Luong, D.X.; Bets, K.V.; Algozeeb, W.A.; Stanford, M.G.; Kittrell, C.; Chen, W.; Salvatierra, R.V.; Ren, M.; McHugh, E.A.; Advincula, P.A.; et al. Gram-scale bottom-up flash graphene synthesis. Nature 2020, 577, 647–651. https://doi.org/ 10.1038/s41586-020-1938-0

Round 2

Reviewer 1 Report

Comments and Suggestions for Authors

accept in present form

Reviewer 2 Report

Comments and Suggestions for Authors

The authors have revised this manuscript based on my comments carefully. Thus I recommend it to be accepted by Nanomaterials.